# Neural Architecture Search Survey: A Computer Vision Perspective

**DOI:** 10.3390/s23031713

**Published:** 2023-02-03

**Authors:** Jeon-Seong Kang, JinKyu Kang, Jung-Jun Kim, Kwang-Woo Jeon, Hyun-Joon Chung, Byung-Hoon Park

**Affiliations:** 1AI Robotics R&D Division, Korea Institute of Robotics & Technology Convergence, Seoul 06372, Republic of Korea; 2Independent Researcher, Seoul 04620, Republic of Korea; 3T3Q Co., Ltd., Seoul 06372, Republic of Korea

**Keywords:** artificial intelligence (AI), deep learning (DL), convolutional neural network (CNN), automated machine learning (Auto-ML), neural architecture search (NAS), computer vision (CV)

## Abstract

In recent years, deep learning (DL) has been widely studied using various methods across the globe, especially with respect to training methods and network structures, proving highly effective in a wide range of tasks and applications, including image, speech, and text recognition. One important aspect of this advancement is involved in the effort of designing and upgrading neural architectures, which has been consistently attempted thus far. However, designing such architectures requires the combined knowledge and know-how of experts from each relevant discipline and a series of trial-and-error steps. In this light, automated neural architecture search (NAS) methods are increasingly at the center of attention; this paper aimed at summarizing the basic concepts of NAS while providing an overview of recent studies on the applications of NAS. It is worth noting that most previous survey studies on NAS have been focused on perspectives of hardware or search strategies. To the best knowledge of the present authors, this study is the first to look at NAS from a computer vision perspective. In the present study, computer vision areas were categorized by task, and recent trends found in each study on NAS were analyzed in detail.

## 1. Introduction

The artificial neural network (ANN) used today is the product of the combined efforts of a number of researchers, including McCulloch et al. [1], who first developed the concept in 1943. The concept of the convolutional neural network (CNN), among the most widely used neural network structures in computer vision (CV) applications, was first introduced using LeNet-5 in a 1989 study by LeCun et al. [2]. Back then, however, due to the lack of computing power of the hardware, CNN was found to be ineffective in dealing with complex and sophisticated tasks, such as object recognition and detection. After quite a while, AlexNet [3] was first introduced at the ImageNet Large Scale Visual Recognition Challenge (ILSVRC) in 2012; it made an impressive debut because the model was able to effectively overcome the limitations of deep learning (DL), making the most of GPU computing. With its unparalleled performance, AlexNet ushered in a new era of DL in earnest [4]. DL has attracted significant attention from researchers thanks to its ability to allow the automatic extraction of valid feature vectors from image data. Performing this automatic extraction requires the tuning of the hyper-parameters of DL. Thus far, extensive research for improved performance has been performed on a wide range of hyper-parameters, such as network structures, weight initialization, activation functions, operators, and loss functions [5,6,7,8,9]. However, the application of DL requires technical expertise and understanding, while incurring a significant amount of engineering time. In attempts to address these issues, automated machine learning (AutoML) has recently garnered significant attention.

The AutoML process is divided into multiple steps, including data preparation, feature engineering, model generation, and model estimation [10]. The first step of AutoML, data preparation, is a process in which data are collected; factors that may negatively affect the learning process, e.g., noise, are removed to train a given model to perform the target tasks. The second step, feature engineering, extracts features from given data to be used in the model. At the same time, feature construction techniques to improve the representative ability of the model, as well as feature selection techniques used to avoid overfitting, are applied in parallel. Conventional feature engineering methods include SURF [11], SIFT [12], HOG [13], the Kalman filter [14], and the Gabor filter. Recently, CNN and recurrent neural network (RNN) applications have been used to extract high-level features. The third step, model generation, is performed to obtain optimum output based on the extracted features. Model generation can be further subdivided into search space, hyper-parameters, and architecture optimization. During this process, model optimization is performed in combination with the model estimation step.

This paper summarizes the basic concepts of neural architecture search (NAS), while providing an overview of previous studies regarding the application of NAS in CV fields. NAS is a research field that encompasses feature engineering, neural networks, architecture optimization, and model estimation, which constitutes AutoML. The search space and search strategy of NAS, as well as major relevant studies, are introduced separately in Section 2.1 and Section 2.2., Section 2.3 introduces the NAS evaluation strategies, and Section 3 will take a close look at the major findings of previous studies on the use of NAS to address various challenges (detection, segmentation, generation, etc.) in addition to classification applications.

## 2. Neural Architecture Search

### 2.1. Search Space

If we simply think of the mechanism of a neural network, it is a graph composed of many operations, and it is a set of operations aimed at transforming from input data to expected output data. Such an expression can be represented by a directed acyclic graph (DAG) as Equation (1) [15].
(1)$$s^{\left(t\right)}=o^{\left(t\right)}\left(s^{\left(t-1\right)}\right)$$
where $s^{\left(t\right)}$ indicates a state at t $s^{\left(t\right)}\in S$ (is equal to the $k^{th}$ node of the neural network), and $s^{\left(t-1\right)}$ and $o^{\left(t\right)}$ represent a previous state and its associated operation $o^{\left(t\right)}\in O$, respectively. O includes various series of operators such as convolution, deconvolution, skip-connection, pooling, and activation functions.

Neural architecture search space consists of a set of candidate operators and a restricted space in which the operators are placed. The operators of the candidate set depend on the task of the neural network. Most NAS research competes for performance through classification challenges in the CV field. In this case, convolution, dilated convolution [16], depth-wise convolution [17], etc., are widely used for candidate set. Search Space has various methods such as Sequential LayerWise Operations [18,19], Hierarchical Brench Connection Operations [20], Global Search space, Cell-Based search space [21], and Memory bank representation [22]. In this article, we discuss the search space divided into global search space and cell-based search space. Figure 1 shows a simple example structure of both methods.

#### 2.1.1. Global Search Space

The global search space is equivalent to finding the series of operators O of the DAG. In other words, it is to determine the operator set with the best performance for a predefined number of nodes between the initial input and the final output. Searching the entire structure has several unavoidable disadvantages. Typically, most deep neural networks used in recent research consist of tens to hundreds of operators. If the global search method is applied to these networks, considerable GPU time will be required.

#### 2.1.2. Cell-Based Search Space

Cell-based search space is based on the fact that the latest and successful artificial neural network structures [23,24,25] consist of repetitions of cells (also known as modules, blocks, units). Typical examples of such cell structures are a residual cell of ResNet [23] and a dense cell of DenseNet [24]. Since the aforementioned neural network structures are flexible in controlling the number of cells, the size of the structure can be determined by optimizing it for a scenario.

NAS research using cell-based search space was introduced in NASNet [21]. NASNet is a structure in which a reduction cell and a normal cell are found through RNN and reinforcement learning, and the final architecture is determined by stacking them several times. In [21], the computational cost and performance was improved compared to the state-of-the-art alternatives (e.g., Inception v3 [26], ResNet [23], DenseNet [24], ShuffleNet [27], etc.).

### 2.2. Search Strategy

As the search space is determined, we need a solution to search for the optimal neural architecture. To automate architecture optimization, traditional techniques up to the latest techniques are being studied. After architecture search, most NAS methods use grid search, random search, Bayesian Optimization, or gradient based optimization for hyperparameter optimization [10]. In this paper, we focus on the search strategies for CV and classify architecture optimization methods into three search strategies and introduce the characteristics of each search technique and the latest research in subsections. Table 1 compares the performance and efficiency of NAS algorithms using the CIFAR-10 and ImageNet dataset.

#### 2.2.1. Evolutionary Algorithm

An evolutionary algorithm (EA) is a metaheuristic optimization algorithm inspired by evolutionary mechanisms in biology such as reproduction, mutation, and recombination. Specifically, an EA finds the optimal solution by repeating the steps of ‘evaluating the fitness of individual’, ‘selecting the individuals for reproduction’, and ‘crossover and mutation operation’. Because this algorithm does not have to assume fitness ideally, it can perform well in approximate solutions for many types of problem.

One of the most popular types of EA is the genetic algorithm (GA). The GA aims to transmit superior genetic information to the next generation by selecting competitive individuals from current generation. Genetic information for NAS includes information about neural network architecture, and each generation is accompanied by a training process to evaluate the performance of neural network architecture. This issue shows the limitation of the EA-based NAS method that requires significant GPU time. The schematic process of GA-based NAS can be summarized as Algorithm 1.
**Algorithm 1** GA-based NAS Algorithm1:  **Input:** the number of generations *T*, the number of individuals in each generation *N*, the dataset *D*;2:  **Initialization:** generating the initial individuals randomly;3:  **For** 1,2,3, …, *T*
**do**;4:  **For** 1,2,3, …, *N* **do**;5:  **Evaluation:** neural network architecture (individual) training for *D_train*;6:  Evaluate the performance of the neural network architecture for *D_test*;7:  **end for**8:  **Selection:** selecting the most suitable individuals for reproduction;9:  **Crossover and Mutation:** reproducing new individuals; 10:**end for**11:**Output:** individuals in the last generation.

In the case of NAS based on the GA, it is important to optimize the structure of the neural network through its genetic representation. Rikhtegar et al. [28] utilized the genetic representation encoded as integers for the number of filters, the size of filters, interconnection between feature maps, and activation functions. Using this encoding method, it is possible to find optimal parameters of neural network architecture, but the number of layers is fixed and it is only possible to search for a pre-defined connection structure. Xie et al. [29] designed a genetic representation focusing on the connection structure of convolutional layers. Their encoding method expressed the validity of the connection between each convolutional layer as a fixed-length binary string. However, contrary to [28], multiscale information cannot be utilized because the number of filters and size of filters are fixed.

#### 2.2.2. Reinforcement Learning

Reinforcement learning (RL), inspired by behavioral psychology, is a method of machine learning in which agents learn in a predefined environment to select actions that seek to maximize the reward obtainable in the current state. Zoph et al. [18] is one of the earliest applications of reinforcement learning for NAS. They argued that Recurrent Neural Network (RNN) can be used as a NAS controller based on the fact that the structure and connectivity of the neural network can be expressed as a variable-length string. The RNN-based controller has the strength to flexibly respond to the architecture structure. In addition, the overall process speed was increased by using the asynchronous parameter updates for the controller and the distributed training for the child network [30]. In the same year, Baker et al. [19] proposed a MetaQNN that combines two techniques with Q-learning.
sensors-23-01713-t001_Table 1Table 1Performance and Efficiency of NAS Algorithms on CIFAR-10 and ImageNet. The ‘S.S’ column indicated the search strategy. The dash (-) indicates that the corresponding information is not provided in the original paper.S.S.ApplicationPerformance (ACC, %) and EfficiencyOpen SourceCIFAR-10ImageNetTop-1GPU Days#GPUsTop-1/5GPU Days#GPUsMResNet [23]93.57--80.62/95.51--https://github.com/KaimingHe/deep-residual-networks, accessed on 30 July 2016DenseNet [24]96.54--78.54/94.46--https://github.com/liuzhuang13/DenseNet, accessed on 11 September 2022EAGeNet#2 [29]92.917-72.13/90.2617--RLMetaQNN [19]93.0810010-

https://github.com/bowenbaker/metaqnn, accessed on 19 July 2017BlockQNN-Connection more filter [31]97.659632(1080Ti)81.0/95.429632 (1080Ti)-BlockQNN-Depthwise, N = 3 [31]97.35ENAS + macro [32]96.130.321---https://github.com/melodyguan/enas, accessed on 2 May 2019ENAS + micro + c/o [32]97.110.451GDDARTS (1st order) + c/o [33]97.001.54(1080Ti)73.3/81.34-https://github.com/quark0/darts, accessed on 12 October 2018DARTS (2nd order) + c/o [33]97.234DARTS+ [34]97.800.41(V100)76.10/92.806.81(P100)-DARTS+ (Large) [34]98.32-Fair-DARTS [35]97.46--75.6/92.603-https://github.com/xiaomi-automl/FairDARTS, accessed on 10 August 2020P-DARTS + c/o [36]97.500.3-75.6/92.600.3-https://github.com/chenxin061/pdarts, accessed on 18 February 2020P-DARTS (large) + c/o [36]97.750.3-Robust DARTS [37]97.05-----https://github.com/automl/RobustDARTS, accessed on 21 July 2020Sharp DARTS [38]98.070.81(2080Ti)74.9/92.200.8--PC-DARTS+ c/o [39]97.430.11(1080Ti)75.8/92.703.88(V100)https://github.com/yuhuixu1993/PC-DARTS, accessed on 3 July 2020


In traditional reinforcement learning, the convergence speed is slow due to over-exploration, and there is a problem if there is convergence to the local minima due to over-exploitation [40]. Baker et al. applied facts that can overcome these two problems with an ε-greedy strategy [41] and experience replay [42]. Zhong et al. [31] also introduced BlockQNN using the Q-learning paradigm with ε-greedy exploration strategy. However, unlike [19], they adopted a cell-based search space, referring to the fact that superior hand-crafted networks such as GoogLeNet [43] and ResNet [23] have the same block stacked structure. In [31], BlockQNN demonstrated advantageous in computational cost and strong generalizability through experiments. Inspired by the concept of ‘weight inheritance’ in [44], Pham et al. introduced ENAS [32], which applies a parameter sharing strategy to child architecture. According to the experimental results of [32], a searching time less than 16 h was required with a single GPU (GTX 1080Ti) on CIFAR-10. This result means that the GPU time is reduced by more than 1000× compared to [18], and it shows considerable efficiency in terms of computational cost.

#### 2.2.3. Gradient Descent (GD)

Since the EA and RL search strategies are based on a discrete neural network space, a lot of search time is required. To solve this problem, studies on how to apply gradient-descent optimization techniques by transforming the discrete neural network space into continuous and differentiable spaces have been published. In 2018, Shin et al. [45] published DAS that solved the NAS as a fully-differentiable-problem, further developing from studies [46,47,48] that partially utilized the continuous domain concept. However, DAS has a limitation of optimizing only specific parameters of an architecture. DARTS [33] published by Liu et al. in 2019 focused on learning the complex operation graph topologies inside the cell constituting the neural network architecture. The DARTS is not limited to a specific neural network architecture, and has been the basis for many studies [34,35,36,37,38] as it can search both CNN and RNN. Liang et al. [34] observed performance collapse as the number of epochs of DARTS increased, and found that this was a result of overfitting due to the number of skip-connections that increased with epochs. They solved the performance collapse problem through DARTS+ [34], which applied the ‘early stopping’ technique, based on the research results that important connections and significant changes were determined in the early phase of training [49,50,51]. Chu et al. proposed Fair-DARTS [35], which applied independence of each operation’s architectural weight to solve the performance collapse problem caused by skip-connection, and eliminated the unfair advantage. Chen et al. [36] proposed progressive DARTS (P-DARTS) to solve the depth gap of DARTS. The depth gap is a problem that occurs in cell-based search methods, and is a performance collapse problem that occurs when different networks operate in the search phase and the evaluation phase. To solve the depth gap problem of DARTS, where the number of cells in the search phase and estimation phase are 8 and 20, respectively, P-DARTS reduced the difference from the estimation phase by gradually increasing the number of cells in the search phase. PC-DARTS [39] proposed by Xu et al. improved memory efficiency through random sampling for operation search and used a larger batch size for higher stability.

### 2.3. Evaluation Strategies

In addition to search space and search strategy, evaluation strategies are very important to improve NAS performance. An efficient NAS performance evaluation strategy can speed up the entire neural architecture search process while reducing the cost of computing resources. The simplest and most popular method, early-stopping [34,52,53,54] was widely used, but search results mostly showed simple and shallow layers. In general, these networks can be fast in terms of speed, but often have poor performance. The technique of parameter sharing is used to accelerate the process of NAS. For example, ENAS [32] constructs a large computational graph to perform a considerable amount of computation so that each subgraph represents a neural architecture and all architectures share parameters. In that way, this method improves the efficiency of NAS by avoiding training sub-models from scratch to convergence. Network morphism based algorithms [55,56] can also inherit the parameters of previous architectures, and single-path NAS [57] uses a single-path over parameterized ConvNet to encode all architectural decisions with a shared convolutional kernel. Another strategy is to use a proxy dataset. This method [58] is used by most existing NAS algorithms and can save computational cost by training and evaluating intermediate neural architectures on a small proxy dataset with limited training epochs. However, it is difficult to expect an architecture search with good performance with this coarse evaluation method.

Recently, various Evaluation Strategies are being studied to overcome the disadvantages of NAS. ReNAS [59] tries to predict the performance ranking of the searched architecture instead of calculating the accuracy as another strategy to accelerate the search process. In 2021, CTNAS [60] completely transforms the evaluation task into a contrastive learning problem. By proposing a method, an attempt is made to calculate the probability that the performance of the candidate architecture is better than the baseline. Most recently in 2022, ERNAS [61] proposes an end-to-end neural architecture search approach and a gradient descent search strategy to find candidates with better performance in latent space to efficiently find better neural structures.

## 3. NAS for CV

### 3.1. Detection

Image detection is a popular technique among the challenges of CV in the industry, and it is a task to find the location of target objects in the input image. In detail, the detection algorithm should solve both the classification problem and the localization problem. There are two types of detection algorithms based on deep learning; one is a two-stage detector and the other is a one-stage detector. The two-stage detector performs object classification and localization separately, but the one-stage detectors do this simultaneously. Representative examples of two-stage detectors include R-CNN [62], SPPNet [63], Faster R-CNN [64], and FPN [65], and one-stage detectors include YOLO [66], SSD [67], and RetinaNet [68].

Likewise, new neural network models for detection have emerged and evolved in various forms year after year. Furthermore, model version upgrades have been consistently performed for improvised performance (e.g., one-stage detectors, including R-CNN [62], Fast R-CNN [69], and faster R-CNN [64], and two-stage detectors, including yolo v1–yolo v7 [66,70,71,72,73,74]). However, to date, almost no NAS methods regarding the object detection network, especially with respect to their efficiency and accuracy, have been studied despite their significant importance.

In early 2019, Ghiasi et al. [75] proposed an FPN search [65] using NAS for object detection. FPN is one of the most widely used model architectures that generates pyramid-type feature representations for object detection. PANet proposed in a study [76] combined the FPN function with extra bottom-up pathways, exhibiting an improved feature representation with respect to low-resolution functions. In many previous studies [77,78,79,80,81,82], a wide range of cross-scale connections or operations to combine multiple features for pyramid-type feature representations were proposed. The major contributions of Ghiasi et al. [75] include the design of a search space that encompasses all possible cross-scale connections for the generation of multiscale feature representations. Their research aimed at finding the “atomic architecture” that can be applied in a repetitive manner in which all inputs and outputs have the same feature level during the entire course of the search. Their results were closely related to those of [18,19,21,83]. The researchers designed a cell (or a layer) to obtain NASNet, a network with state-of-the-art accuracy in ImageNet, using the controller RNN [19] combined with reinforcement learning. The method proposed in a study by Ghiasi et al. [75] was designed based on RetinaNet [68], a simple and efficient framework. In this method, the controller was trained to select the optimum model architecture in a given space using reinforcement learning.

In early studies on NAS [21,84,85], backbone networks designed for general-purpose image classification were employed. Technically, however, classification clearly differs from detection in that classification focuses on finding out what the object is, while detection aims at finding locations and targets. Thus, the use of backbone networks designed for classification is deemed a mere temporary expedient [86]. Furthermore, the performance of an object detector significantly depends on the features extracted from the backbone. Therefore, backbones need to be precisely designed [87]. In a 2019 study, Chen et al. [86] proposed DetNAS, a framework used to design the backbone network of an object detector using NAS. This was the first attempt to implement a backbone search function for object detectors without any proxies. DetNAS allows algorithms to be applied based on the one-shot supernet, which includes all possible networks in the search space. Once the supernet is trained, architectures are explored using the evolutionary algorithm (EA) on the trained supernet, and their performance is evaluated. Based on the results, retraining is performed. In addition to the backbone search proposed in [86], Jiang et al. [88] proposed a two-phase serial-to-parallel architecture search (SP-NAS) framework for the search of flexible task-oriented detection backbones in a 2020 study. The serial searching phase aims to find the sequence of serial blocks with the optimal size and output channels in the feature hierarchy using the Swap-Expand-Reignite search algorithm. Meanwhile, in the parallel searching phase, multiple sub-architectures are combined with the backbones that have been previously searched to obtain more powerful parallel-structured backbones. During iterative training, weight factors are reused to improve the search efficiency and maintain the performance rankings of the networks that have been searched by all training models. In the same year, Yao et el. [89] proposed a structural-to-modular neural architecture search (SM-NAS) framework for object detection. SM-NAS provides a two-step, coarse-to-fine searching strategy for an improved GPU-friendly design of all module-level architectures for a more efficient combination of modules and object detection. The structural-level searching phase aims to find a more efficient combination of different modules, while the modular-level searching phase is a process in which specific modules are allowed to evolve, and the Pareto is inserted into the front of networks with fast operations among all networks. In addition, detection backbones are immediately searched by performing a training search from the start without any pre-trained models or proxy operations. The use of NAS in vision object detection tasks has significantly reduced the amount of manual work in network design because NAS allows for an automatic search of optimal architectures. However, in general, NAS algorithms require an excessive amount of calculation and many GPUs. Furthermore, fast-version NAS algorithms for detection have not been sufficiently studied so far. In a 2020 study, Wang et al. [90] proposed a NAS-fcos method capable of finding the architecture with the best performance within a significantly reduced period of search time. Using this method, the decoder structures of object detectors can be searched while considering the search efficiency. This method, using customized reinforcement learning paradigms, aims to efficiently search not only simple object detectors without anchors, i.e., the prediction heads of Fcos [91], but also feature pyramid networks (FPN). The experimental results obtained using the COCO dataset demonstrated that the method was faster and more efficient than existing NAS-based methods for object detectors, and its compatibility with various backbone architectures confirmed its superior application flexibility. In late 2020, Fang et al. [92] stressed the importance of the backbone search for object detection and segmentation, while putting an emphasis on the need to apply different backbone design principles when performing different tasks. However, backbone pretraining is inevitably a costly process, and a fast network adaptation (FNA++) method can be proposed as a solution to this limitation. In this method, a seed network can be converted into one with a different depth, width, or kernel size using parameter remapping, which adjusts both the architecture and parameters of the seed network. For implementation of architecture-level adaptation, NAS methods were adopted [21,33,83]. The architecture adaptation of FNA++ [92] was found to be much more efficient due to the parameter remapped super network compared to the existing methods for object segmentation or detection, which start the search from the starting point [86,93]. ENAS [32], among the existing methods, proposed the sharing of parameters for reduced search costs, but this sharing strategy may lead to inaccuracy in the evaluation of architectures. In [32,94,95], the sharing of parameters was applied in child models, and this approach was essentially deemed to be parameter remapping. In other studies [57,96], kernel-level parameters were shared in one-shot models. Net2Net [55] proposes a function-preserving transformation in which the parameters of a network are remapped to form a new network with a larger depth and width. This remapping mechanism accelerates the training of the resultant larger network, achieving excellent performance. Compared to Net2Net [55], the remapping mechanism of FNA++ is more flexible and thus compatible with architectures with a wider range of depth, width, and kernel size. This mechanism also allows for data transmission between different tasks while performing architecture and parameter adaptation at low cost, thus helping achieve excellent performance.

In a 2021 study, Yao et al. [97] proposed Joint-DetNAS as a NAS framework for object detection. This technology combined the following three key components: NAS, pruning, and knowledge distillation (KD). First, a general-purpose detector or NAS-searched detector with a lower degree of complexity is used for the search, followed by pretraining using pruning regularization losses to remove redundant parameters. Finally, KD is employed to train the pruned detector. This is how joint optimization is performed. The algorithm used in this approach is composed of two key processes, i.e., student morphism and dynamic distillation. Student morphism is an algorithm used to optimize students’ architectures and remove redundant parameters. Dynamic distillation is an algorithm aimed at finding the optimal matching teacher. Student morphism adopts a weight inheritance strategy, allowing students’ architectures to be updated in a flexible manner and significantly accelerating the search process by fully using the weight factors of previous models. In addition, the integrated progressive shrinking strategy to facilitate dynamic distillation allows the elastic teacher pool to be trained, and thus teacher detectors can be sampled without additional costs in the subsequent searches. In a 2022 study, Viriyasaranon et al. [98] proposed NASGC-CapANet, designed to explore the backbone of object detectors and the feature pyramid representations using NAS and the capsule network. In the study, the NAS-gate convolutional module and the capsule attention module were introduced. Multiple convolution conditions were optimized to overcome the variation of the object scale found in the NAS-gate convolutional module, and the standard convolution was also optimized based on DARTS [33] to reduce the computational costs of backbones. The capsule attention module employs the capsule network’s strong spatial relationship encoding function to generate a spatial attention mask, which puts an emphasis on important features in the feature pyramid while suppressing unnecessary features, to optimize the feature representation and localization function of detectors. Table 2 compares the performance of NAS-based object detectors using the MS COCO dataset, along with information regarding open sources.

### 3.2. Segmentation

Image segmentation is the task of obtaining pixel-level object information from the image data, and is an essential algorithm for visual understanding. Visual understanding is a necessary component for advanced technologies in many other fields such as medical images analysis, autonomous driving applications, and intelligent surveillance systems.

As discussed in Section 3.1, it is also difficult to expect excellent performance in segmentation tasks if image classification models are naively ported. This is because the optimal architecture for segmentation must be able to operate on high-resolution images; however, in image classification, NAS generally adopts transfer learning from lower- to higher-resolution images [21]. In the segmentation process, all significant information is collected at the pixel level, and thus, compared to other vision tasks, this type of task is more sensitive to the resolution of input data. In most previous studies on NAS, a two-level hierarchical design was adopted, allowing an automatic search at the inner cell level only while directly designing the outer network level. This limited search space makes the approach less favorable for dense image prediction, which is sensitive to spatial resolution changes. In an attempt to address this limitation, in a 2019 study, Liu et al. [93] proposed a trellis-like network-level search space, which reflected not only a cell-based search space but also a network-level search space. The architecture found in accordance with the rules proposed by the authors was a combination of down-sampling and up-sampling components, unlike the existing network structure, and the validity and significance of the proposed methodology were experimentally verified in their study.

Previous studies on segmentation, especially aimed at improving the accuracy and convergence speed of compact models, include the works of Liu et al. [100] and Nekrasov et al. [101], which employed knowledge distillation [102]. Later in 2019, Chen et al. [103] proposed a NAS-based approach for segmentation, combined with the concept of distillation for the first time. To be more specific, the researchers unveiled FasterSeg, which skillfully combined the search space of NAS with existing multi-resolution branches, i.e., hand-crafted methods based on human inputs that had already been proven highly effective, thereby allowing real-time inference. Decoupled and fine-grained latency regularization was proposed to address the issue of ‘architecture collapse’ occurring during the latency-constrained search, and at the early stages of the search, a teacher-student co-searching strategy was applied to achieve improved accuracy. This method exhibited superior performance compared to the existing methods in the FPS conditions that can be applied in real-time.

Semantic segmentation is mainly used in the biomedical field to segment and detect boundaries of different anatomical structures in two-dimensional and three-dimensional medical imaging. Given the nature of medical data, which is large-scale, even applying general deep-learning methods requires large amounts of resources. Therefore, significant amounts of resources are also needed for the NAS process, and thus it is important to ensure efficiency in NAS operations. In a 2019 study, Bae et al. [104] proposed RONASMIS, pointing out that it would be difficult to apply the existing NAS methods that had been quite successful in natural image processing, such as DARTS [33] and E-NAS [32] to 3D medical imaging. This method employed a series of techniques to improve the efficiency of search operations, such as micro search, non-reinitializing the child network’s weights, and depth-wise convolution instead of a normal convolution, while putting an emphasis on reducing the training time and the amount of GPU computational power. In the same year, Kim et al. [105] proposed a segmentation neural architecture search framework that was effective in processing high-resolution three-dimensional medical images, which required large amounts of resources. The proposed approach was a cell-based searching operation based on U-Net [25] composed of four types of cells (encoder, reduction, decoder, and expansion). Gumbel-softmax [106] sampling and stochastic bi-level optimization were applied to optimize the complex search space. In a 2020 study, Calisto et al. [107] proposed a novel 2D–3D ensemble network by combining 2D and 3D FCN with feature information to allow automatic adaptation to specific medical segmentation tasks while making use of volumetric information more efficiently. This was the first study that applied the multi-objective NAS system to 3D medical image segmentation. The developed model, also known as AdaEn-Net, has an encoder–decoder structure that automatically adapts to specific image datasets. In the same year, Xu et al. [108] proposed AutoSegNet. Unlike ENAS [32], which includes five operations, the proposed method uses a small search space but significantly increases search efficiency. The down-sampling layer of the network contains a fixed encoder–decoder structure to reduce the input size, and thus pooling operations are removed from the search space and, instead, hybrid dilated convolution [109] operations are added to the search space. Hybrid dilated convolution is a type of dilated convolution without gridding effects. The proposed method is characterized by its ability to learn input features of the previous layers before other receptive fields without grill effects or resolution reduction.

In the semantic segmentation field, many evaluation indexes have been proposed with respect to a range of scenarios. However, despite the widely known and adopted cross-entropy losses and different versions of them, a misalignment between the loss function and the evaluation metrics often leads to degradation in network performance. Furthermore, manually designing loss functions for each metric requires extensive expertise and significant amounts of human resources. In a 2020 study, Li et al. [110] proposed a method of automatically designing loss functions for each metric by searching surrogate losses that were differentiable with respect to each metric. The searched surrogate losses exhibited superior performance in a consistent manner compared to manually designed loss functions.

### 3.3. Generative Adversarial Network (GAN)

The generative adversarial network (GAN) is an algorithm used to generate new sets of data, unlike other algorithms. In the approach, two networks, i.e., the generative network for image generation and the discriminator network for image recognition, compete against each other to grow together. This algorithm is a type of machine learning framework used for unsupervised learning. This technology is used to improve the quantitative and qualitative aspects of data and thus address sparse data issues. It is also used to implement the following four technologies: data synthesis for special conditions based on domain adaptation (snow, rain, nighttime, frost, dust, humidity, etc.); super-resolution to improve the quality of data by increasing the data resolution; semantic synthesis to additionally synthesize specific objects; and image inpainting, in which specific regions of a given image are removed, and the affected parts are processed to look natural. In this section, the necessity and efficient implementation methods of NAS for GAN applications are discussed.

GAN has been highly successful in designing neural architectures, but its implementation generally requires large amounts of time, effort, and expert knowledge. For example, Kerras et al. [111] designed highly complex generators and discriminator backbones to be able to generate high-resolution images more efficiently. To reduce the burden of this network engineering, efficient automatic architecture search techniques for GAN must be ensured. In a 2019 study, Gong et al. [112] proposed an architecture search scheme for GAN, also known as AutoGAN, as part of their first preliminary study on the application of NAS algorithms to GAN. A search space was defined to capture deformations in the GAN architecture, and the RNN controller [18] was employed to guide architecture search operations. Based on the parameter-sharing strategy proposed in a previous study [32], a dynamic parameter resetting strategy was further applied to the search process to increase the training speed. In addition, the multi-level architecture search (MLAS) method, which performs search operations at multiple stages in a bottom-up sequential manner, was introduced. The results confirmed that the performance of AutoGAN was comparable to that of hand-crafted GAN methods at the moment of the tests. Inspired by the success of AutoML in deep compression, Fu et al. [113] developed an AutoGAN-Distiller (AGD) framework by applying AutoML to GAN compression in 2020. The AGD is designed to perform an end-to-end search for efficient generators, and its search operations are based on the original GAN model via knowledge distillation. The developed algorithm was evaluated for performance in the two most frequent GAN tasks, i.e., image translation and super-resolution; the evaluation results confirmed that the algorithm could be generally applied to various GAN models. In the same year, Tian et al. [114] pointed out that although the MLAS method proposed by Gong et al. [112] enhanced stability in policy updates, this progressive formulation might lead to the occurrence of local minimums. In an attempt to address this limitation, the researchers reorganized the GAN architecture search into the Markov decision process (MDP). The MDP formulation facilitated off-policy RL training, improving efficiency in data processing, and as a result, an effective NAS framework, which was six times faster than the existing RL-based GAN search approaches, was proposed. In a 2020 study, Kobayashi et al. [115] employed a multi-objective approach to optimize the network and hyper-parameters for GAN. The researchers pointed out that a naive NAS method might easily lead to an imbalance between the two networks, i.e., the generator and the discriminator [112]. To avoid this problem, they proposed a method of limiting the model size during search and further employed a progressive searching strategy to prevent unstable training operations, such as mode collapses. It was a simple but very efficient search method. In the same year, Gao et al. [116] proposed an AdversarialNAS method. It was a type of NAS approach for GAN and the first method to search both generators and discriminators at the same time in a differentiable manner. The applied search paradigm considered the relevance between the two network architectures and improved the balance between the two. This method did not require additional metrics to be calculated during the search to evaluate the performance of the searched architectures. Therefore, the developed method was found to be very efficient in finding superior generative models in the proposed large search space, proving its performance and superiority. In a 2021 study, Tian et al. [117] proposed a fully differentiable search framework, also known as alphaGAN. The search process was built as equations to solve bi-level minimax optimization problems. For inner-level problems, GAN was trained with the architectures searched with respect to the training dataset. For outer-level problems, a differentiable evaluation metric was utilized to guide the search process toward the pure Nash equilibrium over the validation dataset. StyleGAN2 [118] attempted to expand the search space by integrating the characteristics of backbones, showing that the application of alphaGAN was not limited to specific GAN topologies or small-scale datasets. The developed framework exhibited a speed eight times that of AdversarialNAS [116], which was the result of the most relevant research.

## 4. Discussion

This survey paper provided an extensive review of previous studies on neural architecture search (NAS) as part of AutoML. Notably, with the focus on computer vision (CV) applications, a series of field-specific algorithms based on current NAS techniques for CV, especially those that were expected to contribute to future studies, were introduced and analyzed. Thus far, fewer studies have been focused on detection, segmentation, and GAN applications of NAS for CV than on classification tasks. Furthermore, it is not simple to compare the performance of various NAS methods and approaches with those of existing algorithms reported so far because the NAS method relies on many factors other than the architecture. Most previous studies attempted to evaluate the performance of algorithms of interest using MS-COCO and CIFAR-10, datasets that were mainly used in CV tasks; however, in reality experimental evaluations also involve various other factors, such as hardware, search space, data preprocessing, calculation amount, regularization, and input size. As such, finding an accurate quantitative evaluation method for NAS algorithms for CV, rather than just simply focusing on increasing their performance with respect to each and every task, is deemed to be of great significance.

When analyzing the recent keywords of Computer Vision and Pattern Recognition (CVPR) papers which are published between 2020 and 2022, the keywords of Unsupervised Learning and Self-Supervised Learning are hot topics. Thus, the research related to NAS for CV will gradually be expanded and be advanced. Furthermore, considering the latest technology, we think the search space with gradient descent might be a future direction of development in NAS for CV because it would decrease the search time and be efficient enough to handle the large amount of image data.

## Figures and Tables

**Figure 1 sensors-23-01713-f001:**
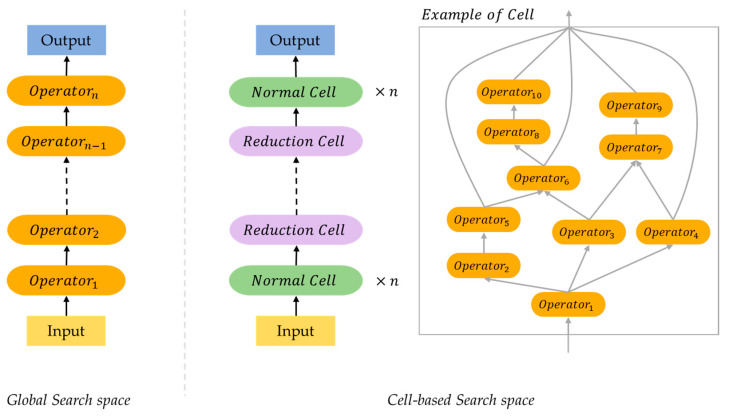
Simplified example of global structured neural architecture and cell structured neural architecture.

**Table 2 sensors-23-01713-t002:** State-of-the-art comparison on MS COCO test-dev for bounding box object detection.

Method	Performance		Open Source
	mAP (%)	FLOPs	
NAS-fpn [75]	45.4	2086.3(B)	https://github.com/open-mmlab/mmdetection, accessed on 16 November 2022
DetNAS [86]	40.2	289.4(G)	https://github.com/megvii-model/DetNAS, accessed on 11 May 2020
SP-NAS [88]	45.6	391.1(G)	https://github.com/mindspore-ai/models/tree/master/research/cv/Spnas, accessed on 21 October 2022
SM-NAS [89]	45.9	162.45(G)	-
NAS-fcos [90]	46.1	361.6(G)	https://github.com/Lausannen/NAS-FCOS, accessed on 10 Jun 2020
FNA++ [92]	36.8	-	https://github.com/JaminFong/FNA, accessed on 12 July 2021
Joint-DetNAS [97]	43.9	153.9(G)	-
NASGC-CapANet [98]	43.8	-	https://github.com/Ewha-AI/Object-Detection_COCO, accessed on 4 March 2021
EfficientDet [99]	41.1	11.0(B)	https://github.com/google/automl/tree/master/efficientdet, accessed on 9 October 2022

## Data Availability

Not applicable.

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
