# Peer review of "Neural Architecture Search Survey: A Computer Vision Perspective"

_sensors, 2023, doi:10.3390/s23031713_

Round 1

Author Response

Response to Reviewer 1 Comments

Neural Architecture Search (NAS) is a field that aims to automate the process of designing neural networks for a particular task. In this paper, the authors present a survey of various NAS methods, covering aspects such as search space design, and search algorithms.

Thank you for your summary.

However, there are some important components that are not adequately addressed in the paper. Specifically, the following areas of NAS are not sufficiently covered:

Point 1. Search Space: NAS search spaces can be divided into four categories: Sequential LayerWise Operations, Hierarchical Branch-Connection Operations, Cell-Based Operations, and Others. Sequential Layer-Wise Operations involve constructing a neural network layer by layer. Hierarchical Branch-Connection Operations involve building a hierarchical network with multiple branches of different depths. Cell-Based Operations use a predefined set of cells or basic blocks that are connected in different ways to form a neural network. The Others category includes any search space methods not covered by the previous three categories.

Response 1: Thank you for your kind comments. The references about NAS search categories which you mentioned have been added in line 85 – 88 as:

“Search Space has various methods such as Sequential LayerWise Operations [18,19], Hierarchical Brench Connection Operations[20], Global Search space, Cell-Based search space[21], and Memory bank representation[22].”

Rather than focusing on the NAS algorithm itself, it was written so that researchers who are in the computer vision (CV) area can explore the NAS algorithm for the current trend in CV. So, the detailed explanation was omitted and replace with references.

Point 2. Search Algorithms: The paper provides a comprehensive overview of various search algorithms for NAS. However, it does not mention important algorithms such as Grid Search, Random Search, and Bayesian Search. Grid Search is a method of exploring all combinations of hyperparameters to find the optimal set, while Random Search involves randomly sampling hyperparameters and evaluating their performance. Bayesian Search is a probabilistic approach that utilizes Bayesian optimization.

Response 2: Thank you for your point. We agree with you about those algorithms which are grid search, random search, and Bayesian search are important algorithms for NAS in general. However, the scopes of this papers focus on the NAS for CV. As shown in Table 1, studies using image datasets (CIFAR-10, ImageNet) as NAS experiments for CV mainly used Reinforcement Learning (RL) and Gradient Descent(GD) methods. Again, the purpose of this paper is to provide useful information for CV research to use NAS, so it is mainly focused on recent papers on RL and GD methods. Of course, we added some explanation about this in line 116-118 as:

“After architecture search, most NAS methods use grid search, random search, Bayesian Optimization, gradient based optimization for hyperparameter optimization [10]. In this paper, we focus on the search strategies for CV and …”

Point 3. NAS Evaluation Strategies: The paper discusses several NAS evaluation strategies, such as training from scratch, partial training, and proxy task performance. However, it does not provide a detailed description of each method or the trade-offs and limitations associated with them. It also does not mention other evaluation strategies such as parameter weight sharing and prediction-based approaches. Despite these gaps in coverage, the paper remains a useful resource for researchers and practitioners interested in NAS.

Response 3: Thank you for your comment. As you mentioned, NAS Evaluation Strategies are very important to the success of NAS. In line 67, 203-221, “2.3. Evaluation Strategies section” has been added. The characteristics of each of the recent evaluation strategies were summarized and written.

Reviewer 2 Report

This article is well-written and provides a comprehensive, clear, and appropriate view in regard to the NAS' application in CV. However, the reviewer still suggests some modifications being made before its acceptance.

The article is indeed lack of a part of necessary content, that is, the perspective of the future directions as for the NAS' in CV. In other words, in view of these existing NAS technique, what do the authors think is the next development direction? It should be argued in a separate section. The authors are suggested to add a section of elaboration before the conclusion.

Author Response

Thank you very much for your kind comments. We added paragraph in the discussion section (line 510-516) as “When analyzing the recent keywords of Computer Vision and Pattern Recognition (CVPR) papers which are published between 2020 and 2022, the keywords of Unsupervised Learning and Self-Supervised Learning are hot topics. Thus, the research related to NAS for CV will gradually be expanded and be advanced. Also, considering the latest technology, we think the search space with gradient descent might be a future direction of development in NAS for CV because it would be decreasing the search time and be efficient to handle the large amount of image data.”

Round 2

Reviewer 1 Report

The manuscript has undergone modifications in response to my feedbacks and comment. The contributors have supplied justifications for the remarks and They have provided explanations and have added an additional section to the manuscript. Nevertheless, the recently incorporated segment exhibits inadequate representation of the pertinent literature within the domain, and necessitates the incorporation of additional references.

Overall, I am weakly accepting the paper with the suggestion that they address this issue in their revisions.

Author Response

Response to Reviewer Comments

The manuscript has undergone modifications in response to my feedbacks and comment. The contributors have supplied justifications for the remarks and They have provided explanations and have added an additional section to the manuscript. Nevertheless, the recently incorporated segment exhibits inadequate representation of the pertinent literature within the domain, and necessitates the incorporation of additional references.

Overall, I am weakly accepting the paper with the suggestion that they address this issue in their revisions.

Response: Thank you very much for your kind comments. Your comments and suggestions have improved manuscript a lot. The newly added evaluation strategies section has been modified and supplemented (In lines 203-230). Also, we added some references and incorporated the papers described in other sections. Thank you.